# 2023 Canadian Colposcopy Guideline: A Risk-Based Approach to Management and Surveillance of Cervical Dysplasia

Karla Willows [1,*], Amanda Selk [2,3], Marie-Hélène Auclair [4], Brent Jim [5], Naana Jumah [3,6], Jill Nation [7], Lily Proctor [8], Melissa Iazzi [9] and James Bentley [1]

[1] Nova Scotia Cancer Centre, Division of Gynecologic Oncology, Dalhousie University, Halifax, NS B3H 4R2, Canada

[2] Women's College Hospital, Toronto, ON M5S 1B2, Canada

[3] Department of Obstetrics and Gynecology, University of Toronto, Toronto, ON M5S, Canada

[4] Department of Obstetrics and Gynecology, Division of Gynecologic Oncology, Hôpital Maisonneuve-Rosemont, CIUSSS de l'Est de l'Île de Montréal, Montréal, QC H1T 2M4, Canada

[5] Division of Gynecologic Oncology, Allan Blair Cancer Centre, University of Saskatchewan, Regina, SK S7N 5A2, Canada

[6] Department of Obstetrics and Gynecology, Northern Ontario School of Medicine, Thunder Bay, ON P7B 5E1, Canada

[7] Division of Gynecologic Oncology, Cumming School of Medicine, University of Calgary, Calgary, AB T2N 1N4, Canada

[8] Department of Obstetrics and Gynecology, Division of Gynecologic Oncology, University of British Columbia, Vancouver, BC V6T 1Z4, Canada

[9] The Society of Gynecologic Oncology of Canada (GOC), Ottawa, ON K1H 8K3, Canada

* Correspondence: karla.willows@nshealth.ca

**Abstract:** This guideline provides evidence-based guidance on the risk-based management of cervical dysplasia in the colposcopy setting in the context of primary HPV-based screening and HPV testing in colposcopy. Colposcopy management of special populations is also discussed. The guideline was developed by a working group in collaboration with the Gynecologic Oncology Society of Canada (GOC), Society of Colposcopists of Canada (SCC) and the Canadian Partnership Against Cancer (CPAC). The literature informing these guidelines was obtained through a systematic review of the relevant literature via a multi-step search process led by information specialists. The literature was reviewed up to June 2021 with manual searches of relevant national guidelines and more recent publications. Quality of the evidence and strength of recommendations was assessed using the Grading of Recommendations Assessment, Development, and Evaluation (GRADE) framework. The intended users of this guideline include gynecologists, colposcopists, screening programs and healthcare facilities. Implementation of the recommendations is intended to promote equitable and standardized care for all people undergoing colposcopy in Canada. The risk-based approach aims to improve personalized care and reduce over-/under-treatment in colposcopy.

**Keywords:** colposcopy; human papillomavirus; HPV; cervical cancer; guideline

## 1. Introduction

In 2020, the World Health Organization (WHO) announced a global strategy towards the elimination of cervical cancer as a public health problem by reducing the incidence of cervical cancer to less than 4 per 100,000 [1]. Most cervical cancer is human papillomavirus (HPV)-related and preventable. Over the past 40 years, secondary prevention with screening and treatment of pre-invasive lesions has contributed to significantly reduced incidence and mortality. Primary prevention with HPV vaccination is expected to further reduce the incidence. Despite this, we still see approximately 1450 new cases and 380 deaths from cervical cancer per year in Canada [2]. In line with the WHO recommendations, the Canadian Partnership Against Cancer (CPAC) has proposed an action

plan for the elimination of cervical cancer in Canada by 2030 [3]. Priorities of this plan include the implementation of primary HPV screening and improved follow-up of abnormal screening results.

Primary HPV screening is superior to cervical cytology in detecting pre-invasive disease and is the topic of a complimentary guideline [4]. Once high-risk HPV (HR-HPV) is detected, reflex cytology is used to triage the person's immediate and 5-year risk of a high-grade squamous intraepithelial neoplasia or cancer and the need for colposcopy. A risk-based approach, presented in the 2019 American Society of Colposcopy and Cervical Pathology (ASCCP) Risk-Based Management Consensus Guidelines for Abnormal Cervical Cancer Screening Tests and Cancer Precursors [5] represents a paradigm shift from using results-based algorithms to using risk-based management based on a combination of current and past HR-HPV and cytology results. HPV testing in colposcopy can identify those at lower risk for histologic high-grade squamous intraepithelial neoplasia or cancer, thereby reducing the risk of invasive tests and over-treatment, as well as informing subsequent screening.

The following guideline is presented in the context of primary HPV-based screening for cervical cancer, whereby a person's HPV status is known on entry to colposcopy. The purpose of this guideline is to provide guidance on how to incorporate high-risk HPV (HR-HPV) testing and results within our current colposcopy framework with an emphasis on risk-based management. In Canada, each province and territory administers and delivers most health services independently and needs to consider their own populations and resources. This guideline is not meant to supersede local recommendations; rather, it is meant to provide evidence to advocate for equitable resources and standardized care for all people undergoing colposcopy in Canada.

## 2. Methods

Guideline committee members were recruited to participate in the development of the document with consideration given to including colposcopists from a range of Canadian provinces and practice settings (for example, urban and rural, general gynecologists and gynecologic oncologists). The committee convened in the fall of 2020 and developed a series of objectives to guide the document.

The literature informing the guideline was located using a multi-step search process led by an information specialist. First, a search for existing clinical practice guidelines related to colposcopy for cervical cancer was designed in Ovid MEDLINE All and executed on 12 May 2021. The search combined search terms for colposcopy, cervical cancer and the Canadian Agency for Drugs and Technology in Health guidelines filter for Ovid MEDLINE (REF: https://www.cadth.ca/strings-attached-cadths-database-search-filters (accessed on 10 May 2021). No date or language filters were applied to the search. The search strategy is reported in full in Supplementary Table S1. This search was followed, on 7 June 2021, by a search for primary studies published since 2018, the date of the literature search cut-off for the 2019 ASCCP risk-based colposcopy guidelines. This search was identical to the first, with the guideline filter removed and the date limit added. A second search, specific to the objectives on equity in colposcopy, was designed in Ovid MEDLINE All and executed on 7 February 2022. This search strategy is reported in full in Supplementary Table S2.

The results of both searches were screened at the title/abstract level by two independent reviewers. Records remaining to be assessed at the full-text level were then reviewed in duplicate by four pairs of reviewers, with each pair focusing on specific objectives, defined a priori by the guideline committee. Hand-searching of references from relevant citations was also performed.

Data relevant to the specific objectives were extracted, along with levels of evidence for each citation. Literature summaries were generated for each objective. Recommendations were developed and graded based on the quality of evidence available using the Grading of Recommendations, Assessment, Development and Evaluations (GRADE) framework; this is shown in Supplementary Table S3. Where evidence was limited, expert

consensus recommendations were generated by discussion of the guideline committee members. These recommendations, along with summaries of the supporting evidence, are presented here.

## 3. Results

### 3.1. The Lower Anogenital Squamous Terminology (LAST) for HPV-Related Lesions of the Cervix

Recommendations:

- Histopathology should be reported using the two-tiered terminology described by the LAST Project: LSIL for CIN1 and HSIL for CIN2/CIN3 (conditional, moderate).
- p16 immunohistochemistry may be used to upgrade CIN2 to CIN3. P16 should not be used to upgrade morphologically appearing CIN1 (strong, high).

The Lower Anogenital Squamous Terminology (LAST) Standardization Project for HPV-Associated Lesions, published in 2012, recommends harmonizing cytology and pathology into two-tier reporting [6]. CIN 1 is replaced with the term LSIL and CIN 2/3 is replaced with HSIL. There is debate regarding the use of CIN 2 as there is poor inter-rater reliability among pathologists [7]. In Canada, we have a hybrid of reporting methods, with some pathologists reporting LSIL or HSIL on histology specimens while others continue to report using the CIN 1–3 terminology. For the purposes of this guideline, if CIN2 or CIN3 is reported on histology, then follow the HSIL pathway. If CIN1 is reported on histology, then follow the LSIL pathway. We will continue to refer to CIN 2 as a category as it continues to be published in studies and is especially relevant in discussions of conservative management (see Sections 3.6 and 3.11.1). Acceptable abbreviations and terminology are listed in Table 1.

**Table 1.** Cervical screening and colposcopy terminology.

| **HPV-Related Abbreviations** | |
|---|---|
| HPV | Human papillomavirus |
| HR-HPV | High-risk HPV as identified on HPV genotyping |
| HPV 16/18 | HPV 16 and/or 18 |
| Positive HPV test | HPV test showing high-risk HPV types on genotyping |
| HSIL+ | HSIL or cervical cancer |
| VaIN | Vaginal intraepithelial neoplasia |
| **2014 Bethesda System for Reporting Cervical Cytology [8]** | |
| Normal | Negative for intraepithelial lesion and malignancy |
| LSIL | Low-grade squamous intraepithelial lesion |
| ASCUS | Abnormal squamous cells of undetermined significance |
| ASC-H | Abnormal squamous cells cannot rule out high-grade dysplasia |
| HSIL | High-grade squamous intraepithelial lesion |
| AIS | Adenocarcinoma in situ |
| AGC | Abnormal glandular cells |
| AGC-NOS | Abnormal glandular cells, not otherwise specified |
| AGC-N | Abnormal glandular cells, favoring neoplasia |
| **Cervical Intraepithelial Neoplasia Naming System for Cervical Pathology [9]** | |
| CIN 1 | Cervical intraepithelial neoplasia 1 |
| CIN 2 | Cervical intraepithelial neoplasia 2 |
| CIN 3 | Cervical intraepithelial neoplasia 3 |
| **Colposcopy Terminology** | |
| CKC | Cold knife conization |
| ECC | Endocervical curettage |
| LEEP | Loop electrosurgical excisional procedure |
| LLETZ | Large loop excision of the transformation zone |

Guidance for the use of p16 immunochemistry is published in the LAST project. In general, when the morphologic interpretation of hematoxylin and eosin (H & E) staining is suggestive of HSIL, then positive p16 immunochemistry is confirmatory [6]. It is important that p16 is not used to upgrade morphologically appearing LSIL.

### 3.2. Risk-Based Entry to Colposcopy

Recommendations:

- People with a positive HPV screening test should undergo HPV genotyping and reflex cytology before referral to colposcopy (strong, high).
- People with HPV 16/18 should be referred to colposcopy (strong, high).
- People with HPV 'other' ASCUS or LSIL should have HPV testing repeated at 12 and 24 months, only referred to colposcopy if they meet other criteria or have persistent HPV "other" at 24 months (conditional, moderate).
- People with HPV-positive HSIL, ASC-H, AGC, AIS or cytology suspicious for invasive cancer should be referred directly to colposcopy, regardless of HPV genotype (strong, high).
- People with immunocompromise with any HR HPV should be referred to colposcopy (conditional, low).

Primary HPV-based screening with reflex cytology allows for risk stratification based on immediate and future risk of histologic HSIL+. Large population-based studies, most notably data from the Kaiser Permanente Northern California (KPNC) cohort, have examined the risk of histologic HSIL based on combined HPV and cytology results, contributing to our understanding of these risk profiles [10,11]. Based on American data, the 2019 ASCCP guidelines define a general risk-based entry to colposcopy threshold of 4% [12]. The risk-based threshold for entry into colposcopy should take into account the number needed to be seen in colposcopy to detect one case of actionable histology (i.e., HSIL, AIS or cancer). This number may vary by jurisdiction. Therefore, we acknowledge that this threshold may need to be adapted by individual jurisdictions to reflect their own populations, screening data and resources. For the purpose of this guideline, an immediate risk of actionable histology of 5% or greater has been determined as an appropriate threshold for immediate referral to colposcopy. For further information and discussion on entry to colposcopy, please see the complementary guideline 'A Canadian guideline on the management of a positive HPV test and guidance for specific populations' [4].

Tabulated risks of various combinations of cytology and HR-HPV results are presented in Table 2. All people with a positive HR-HPV screening test should undergo HPV genotyping and reflex cytology before referral to colposcopy. People who test positive for HPV16/18, regardless of cytology, should be referred to colposcopy. People with HPV 'other' ASCUS or LSIL should have HPV testing repeated at 12 and 24 months, and only be referred to colposcopy if meet other criteria or have persistent HPV 'other' at 24 months. People with HPV positive ASC-H, HSIL, AGC, AIS or cytology suspicious for invasive cancer should be referred directly to colposcopy, regardless of HPV genotype. People with immunosuppression, as defined below (Section 3.11.3), are at greater risk for histologic HSIL [13] and should therefore be referred to colposcopy with any HR-HPV result, regardless of genotype or cytology, although reflex cytology is recommended to further stratify their risk.

**Table 2.** Immediate-risk HSIL+ based on primary HPV-based screening and reflex cytology results.

| | HPV | | | |
|---|---|---|---|---|
| Cytology | Pos HR-HPV (Any) | Pos HPV 16 | Pos HPV 18 | Pos HPV Other |
| Normal | 3.4% [10] | 5.3% [10] | 3% [5] | 2% |
| ASCUS | 4.4% [11] | 9% [10]–12.9% [14] | 5% [14] | 2.7% [14]–4.4% [11] |
| LSIL | 4.3% [11] | 11% [10] | 3% [5] | 4.3% [5,11] |
| ASC-H | 26% [5,11] | 28% [5,10] | 15% [10] | 26% [5,11] |
| HSIL | 49% [5,11] | 60% [5,10] | 30% [5,10] | 49% [5,11] |

*3.3. The Initial Colposcopic Exam and Documentation*

Recommendations:

- The transformation zone should be assessed, and the type should be documented (strong, high).
- International Federation of Cervical Pathology and Colposcopy (IFCPC) terminology is recommended for documenting colposcopic findings (strong, high).
- Targeted biopsies of lesions are recommended. In the setting of HSIL, or positive HPV16/18, where colposcopic impression is normal, any area of acetowhitening, metaplasia or uncertainty should be biopsied (strong, high).
- Endocervical curettage and endometrial biopsies are contraindicated in pregnancy (strong, high).
- Endocervical curettage is recommended with: (i) a type 3 transformation zone, (ii) HSIL/ASC-H cytology when no lesion is identified, (iii) AGC/AIS cytology, (iv) when excisional treatment has positive margins and (v) in people over age 45 with HPV 16 (conditional, moderate).
- Endometrial sampling is recommended in those 35 years and older for all categories of AGC/AIS or atypical endometrial cells on cytology. Endometrial sampling is also indicated in those under 35 with increased risks of endometrial cancer (obesity, chronic anovulation or abnormal uterine bleeding) or atypical endometrial cells on cytology in those of any age (strong, moderate).
- For pain management for routine exam and cervical biopsies, thorough pre-procedure counseling and non-pharmacologic methods are recommended. Oral analgesics may be considered. Topical and injected analgesics are not recommended for routine exam and biopsies of the cervix (strong, low).

When the cervix is first examined, cervical mucus is removed and gross examination is performed. Using a colposcope for visualization, 3–5% acetic acid is applied to the cervix, and the transformation zone is then visualized. The transformation zone type is noted (Figure 1) and any lesions identified are described using the International Federation for Cervical Colposcopy and Pathology (IFCPC) terminology as reviewed below [15]. Some colposcopists also apply Lugol's Iodine and then re-examine the cervix.

When abnormal colposcopic findings are identified, IFCPC recommends documenting both the location of the lesion, inside or outside the transformation zone and location of the lesion by clock position, and the size of the lesion, by the number of cervical quadrants the lesion covers and the size of the lesion in the percentage of the cervix affected. Further findings are categorized as grade 1 (minor) or grade 2 (major), suggestive of low- versus higher-grade lesions, respectively. Findings suspicious for invasion should also be described (Table 3).

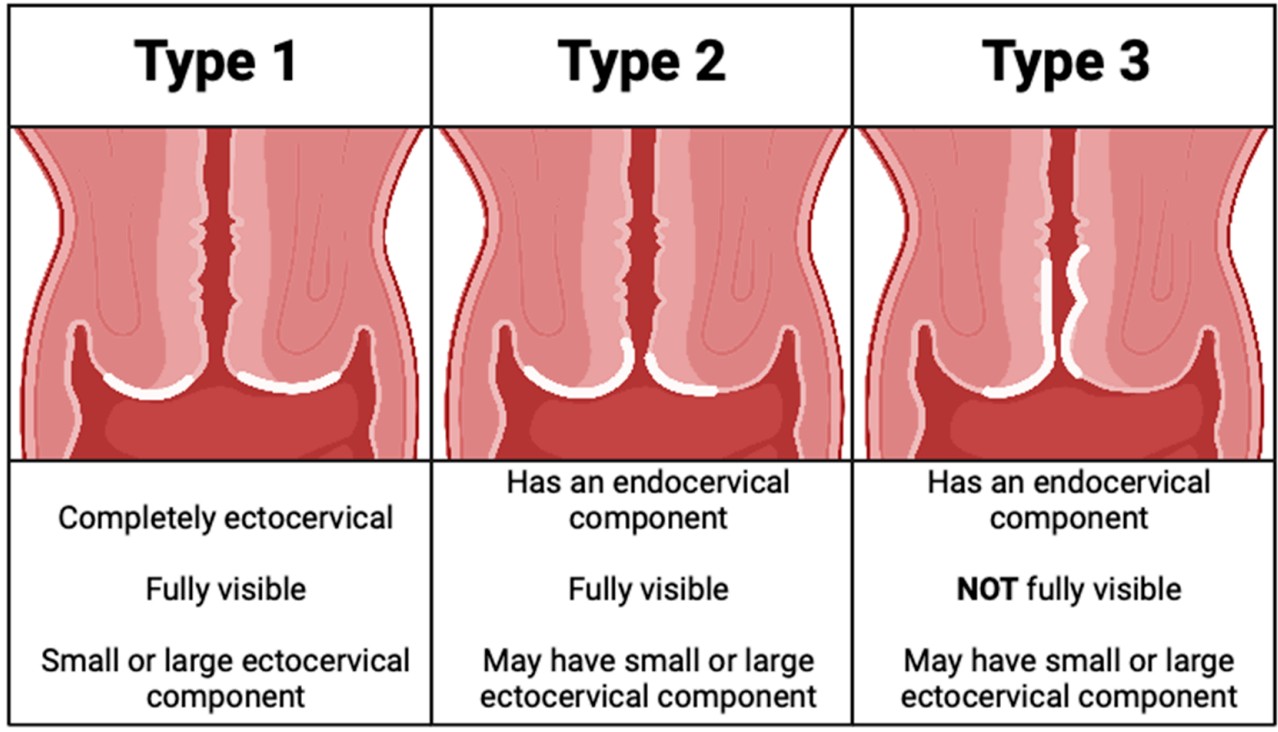

**Figure 1.** Transformation zones [16].

**Table 3.** Findings at the time of the colposcopic exam.

| General Assessment: | Squamocolumnar Junction Visibility: Completely Visible, Partially Visible, Not Visible Transformation Zone Types 1,2,3 (Figure 1) |
| --- | --- |
| Normal Findings: | Original squamous epithelium: mature or atrophic columnar epithelium, ectopy, metaplastic squamous epithelium, nabothian cysts, crypt (gland) openings, deciduosis in pregnancy |
| Grade 1/Minor Findings: | Thin aceto-white epithelium; irregular, geographic border<br>Fine mosaic, fine punctation |
| Grade 2/Major Findings | Dense aceto-white epithelium, rapid appearance of acetowhitening, cuffed crypt (gland) openings, coarse mosaic, coarse punctuation, sharp border, inner border sign, ridge sign |
| Findings Suspicious for Invasion | Atypical vessels, fragile vessels, irregular surface, exophytic lesion, necrosis, ulceration (necrosis), tumor/gross neoplasm |

Targeted biopsy of all discrete lesions is recommended. Studies have shown that taking two or more biopsies improves the sensitivity of the exam and increases the detection of histologic HSIL [17]. Colposcopic-directed biopsies are less accurate in postmenopausal people, and when a type 3 transformation zone is present [18]. Random biopsies are not recommended. However, in the setting of HSIL, or positive HPV16/18, where colposcopic impression is normal, any area of acetowhitening, metaplasia or uncertainty should be biopsied [19].

Endocervical curettage (ECC) is contraindicated in pregnancy. ECC is generally not required with a type 1 transformation zone. ECC is recommended for type 3 transformation zones, HSIL/ASC-H cytology when no lesion is identified, AGC/AIS cytology and on post-treatment follow-up when excisional treatment has positive margins. Furthermore, ECC should be considered in people over age 45 with HPV 16 [20,21]. As people age, the transformation zone recedes up the endocervical canal. In people over 45 years of age, data show that biopsy alone can miss pathology in over 40% of cases [21].

Endometrial sampling is contraindicated in pregnancy. Endometrial sampling is recommended in those 35 years and older for all categories of AGC/AIS cytology, as well as when atypical endometrial cells are found on cytology. Endometrial sampling is indicated in those under 35 with increased risks of endometrial hyperplasia or cancer (obesity, chronic anovulation, abnormal uterine bleeding, etc.) and when atypical endometrial cells are found on cytology in those of any age [22].

Pain management should be a consideration during all colposcopy exams and procedures. Significant discomfort may impede the exam and can contribute to loss of follow-up. Pain during colposcopy is multi-factorial, influenced by physical factors, psychological factors and social factors [23]. Thorough counseling prior to initiating the exam, outlining steps and anticipated duration, can set expectations and help reduce anxiety that contributes to pain. Other non-pharmacologic techniques that have been suggested include: relaxation, guided imagery, distraction and having a support person present [23]. Music and forced cough during biopsy are simple strategies that have shown to be effective in reducing pain and anxiety at the time of the colposcopy exam [24–28]. Data on oral analgesia (NSAIDs, opioids) prior to or after the colposcopy exam are lacking and mixed [29,30]; however, these may be considered in addition to non-pharmacologic techniques.

Several studies have shown that application of topical analgesics to the cervix prior to biopsy and manipulation are ineffective compared to placebo [29,31–33]. A systematic review and meta-analysis by Mattar et al. in 2019 on the use of local anesthetic for pain relief during colposcopic-guided biopsy included 11 randomized controlled trials. They showed that while local anesthetic reduced pain with biopsy, there was no significant improvement in post-procedural pain, pain on endocervical curettage, pain expectancy or overall pain scores [34]. They concluded that at present there is insufficient evidence to recommend local anesthetics for routine colposcopy exams and cervical biopsies. In contrast, local anesthetic remains essential for ablative and excisional treatments of cervical HSIL/AIS (see Section 3.7), and for biopsy of the vulva.

### 3.4. Low-Grade Referral Pathway (Figure 2)

Recommendations:

- After initial colposcopy assessment, those with normal or LSIL histology can be discharged from colposcopy (strong, moderate).
- Where HSIL is identified on histology, excisional procedure is recommended (strong, high).

The majority of abnormal Pap tests are low-grade. Most of these will spontaneously resolve, especially in the young, previously vaccinated cohort [35]. The purpose of cervical screening and colposcopy is to identify and treat histologic high-grade abnormalities (HSIL, suspected carcinoma) and glandular lesions.

The low-grade referral pathway (Figure 2) addresses people referred to colposcopy with HPV-positive ASCUS and LSIL cytology. Immediate risk of histologic HSIL+ in this population, regardless of HPV genotype, is approximately 4.5% [11]; risk is substantially higher, 9–13%, if HPV-16 is confirmed [10]. The following recommendations balance the risk of subsequent histologic HSIL with the risk of over-treatment in colposcopy.

After initial colposcopy assessment, if no histologic HSIL is identified, people can be discharged from colposcopy with recommendations for 12-month HPV-based screening with their primary care provider. This is in line with the 2019 ASCCP risk-based guidelines [5] and is based on data showing the histologic HSIL+ 3-year risk of HPV-positive ASCUS and LSIL with histology that is normal or LSIL is 2.2% and 1.8%, respectively [36]. These people can then be followed based on the post-discharge pathway (Section 3.10; Figure 7).

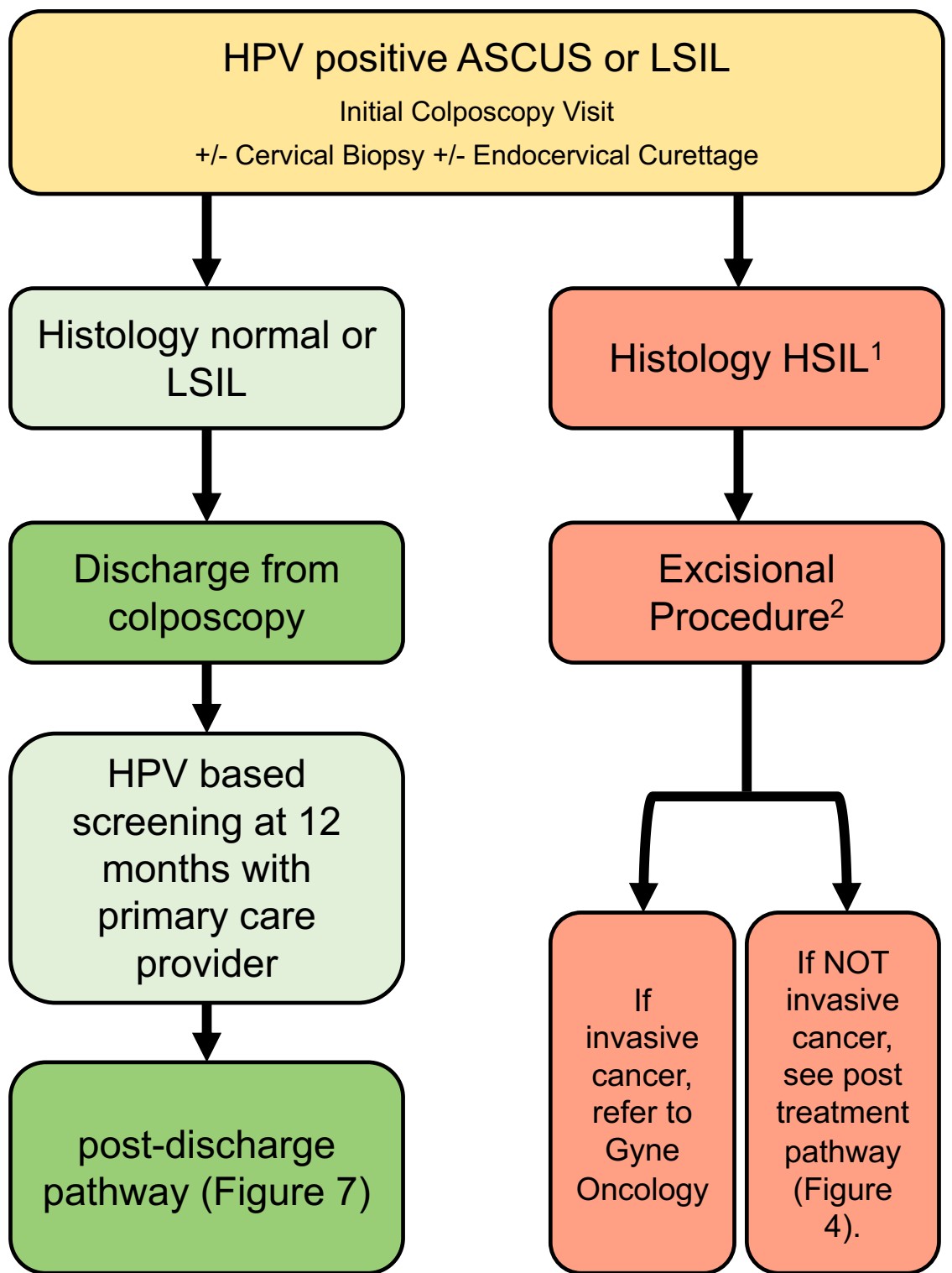

<sup>1</sup> For conservative management of CIN2 identified in patients under the age of 30, please see Figure 4.

<sup>2</sup> Laser ablation may also be used to treat histologic HSIL when specific criteria are met. See section 3.7 of the guideline.

**Figure 2.** Low-grade referral pathway (HPV-positive ASCUS or LSIL). The low-grade referral pathway addresses persons referred to colposcopy with HPV-positive ASCUS and LSIL cytology.

People with histologic HSIL identified at the time of the initial colposcopy assessment should undergo excisional treatment (Section 3.7); they subsequently follow the post-treatment pathway (Section 3.8; Figure 5). Conservative management of histologic HSIL should only be considered in cases of confirmed CIN2 in people under the age of 30 (Section 3.6; Figure 4) or when HSIL is identified in pregnancy (Section 3.11.2).

*3.5. High-Grade Referral Pathway (Figure 3)*

Recommendations:

- People with evidence of histologic HSIL should undergo an excisional procedure (strong, high).
- In cases where no lesion is identified following referral for HPV-positive cytologic ASC-H/HSIL, review by an experienced cytopathologist should be considered (conditional, moderate).
- In cases where no cervical lesion is identified following referral for HPV-positive cytologic ASC-H/HSIL, VAIN must be ruled out by colposcopy (conditional, low).
- In cases of discordance, where no histologic HSIL is confirmed, management depends on transformation zone type and referral cytology (conditional, moderate).
- For a type 3 transformation zone, excisional procedure is recommended (conditional, moderate).
- For a type 1 or 2 transformation zone, the preferred management for ASC-H referral cytology is surveillance. For HSIL referral cytology, excisional procedure can be considered (conditional, moderate).
- If surveillance without treatment is undertaken, people should remain in colposcopy at 6-month intervals with HPV testing at annual intervals until HPV is negative on two consecutive tests and histology remains normal or LSIL (conditional, moderate).
- If HPV remains positive despite negative colposcopy, people should remain in colposcopy for surveillance at 12-month intervals until they meet the above criteria for discharge (conditional, moderate).
- If, during surveillance, there is evidence of cytologic or histologic HSIL, excisional procedure is recommended (strong, high).

The high-grade referral pathway (Figure 3) addresses people referred to colposcopy with HPV-positive ASC-H and HSIL cytology. Immediate risk of HSIL+ in this population ranges from 15–28% for referral cytology of ASC-H and from 30–60% for HSIL cytology [5,10,11]. People with confirmed histologic HSIL should undergo excisional procedure to rule out cancer and treat their pre-invasive lesion.

In cases of discordant cytology, histology and colposcopy findings, the presence and type of oncologic HPV infection, adequacy of colposcopy and transformation zone type need to be considered. For people referred with HPV-positive high-grade cytology, if colposcopy is adequate and no lesion is found, a detailed examination of the endocervix and vagina should be undertaken [37]. Random biopsies of the transformation zone can be considered [19,38]. When colposcopy and biopsies are negative for histologic HSIL after abnormal cytology, it does not mean that HSIL is not present; however, it is unlikely that an occult malignancy has been missed [39]. In these cases, cytopathology review is recommended prior to decision to treat [5,16,40–42].

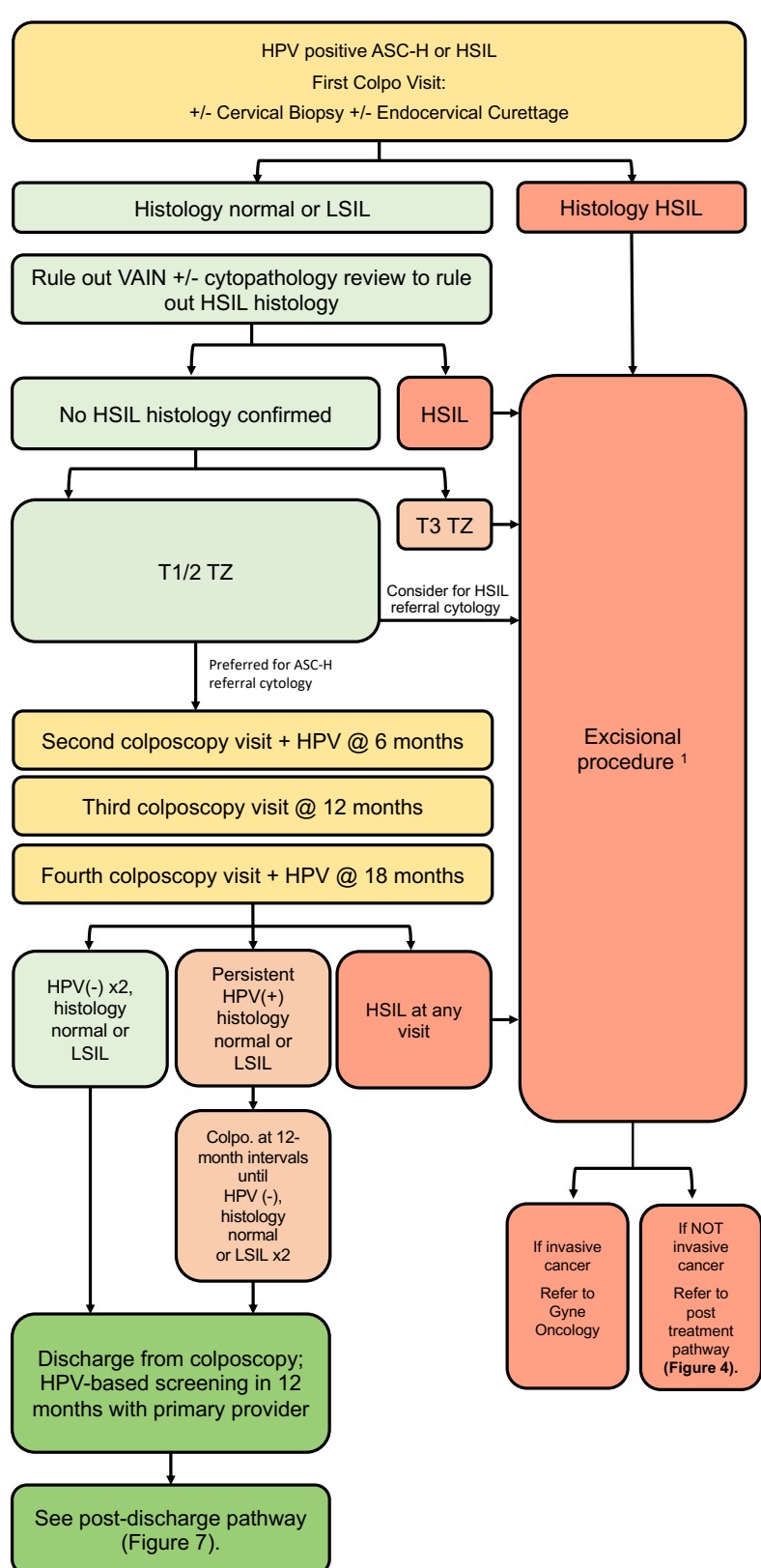

**Figure 3.** High-grade referral pathway (HPV-positive ASC-H or HSIL). The high-grade referral pathway addresses persons referred to colposcopy with HPV-positive ASC-H and HSIL cytology.

For persons where high-grade discordance persists after cytopathology review (i.e., high-grade cytology with normal or LSIL histology), management depends on transformation zone type and referral cytology. All people with HPV-positive high-grade referral cytology (HSIL or ASC-H) with a type 3 transformation zone and otherwise negative colposcopy exam are recommended to undergo an excisional procedure. For those with HSIL referral cytology and a type 1 or 2 transformation zone, an excisional procedure can be considered, whereas close surveillance is preferred for ASC-H referral cytology. This is based on risk estimates that show the 1-year risk of CIN3+ is higher for discordance involving cytologic HSIL/histologic LSIL at 3.9% compared to cytologic ASC-H/histologic LSIL at 1.4% [11]. Those who undergo an excisional procedure in colposcopy should follow the post-treatment algorithm (Section 3.8; Figure 5).

For those who do not undergo an excisional procedure, close surveillance is required with colposcopy at 6-month intervals and HPV testing at annual intervals. If HSIL cytology persists or is identified at any follow-up visit, an excisional procedure is recommended. Once HPV is negative for two consecutive annual tests, and colposcopic impression/histology remain normal or LSIL, people can be discharged back to their primary care provider for repeat HPV-based screening in 12 months (Section 3.10; Figure 7). If HPV is persistently positive (regardless of HPV genotype), people should remain in colposcopy at annual intervals until HPV is negative on two consecutive tests or until histologic HSIL is identified, at which point they undergo an excisional procedure.

*3.6. Conservative Management of CIN2 in People <30 Years Old Where Fertility Is a Concern (Figure 4)*

Recommendations:

- In people under the age of 30, where childbearing considerations outweigh the risk of pre-invasive or invasive disease, and where a pathologic distinction between CIN2 and CIN3 can be reliably made, a conservative approach may be undertaken (conditional, moderate).
- In these cases, review by an expert cytopathologist should rule out CIN3 (conditional, high).
- Findings of CIN2 in a young person with a type 3 transformation zone or CIN2 identified on endocervical curettage should undergo excisional procedure (conditional, moderate).
- Findings of CIN2 in a young person with a type 1 or 2 transformation zone may be managed conservatively for childbearing considerations; surveillance should include colposcopy at 6-month intervals and HPV testing at annual intervals for 3 years to allow these young people to resolve their HPV infections (conditional, moderate).
- If CIN2 persists, continue colposcopy at 6-month intervals with HPV testing annually. People under 30 with persistent CIN2 > 36 months or CIN3 at any colposcopy visit should have an excisional procedure (conditional, low).
- People under the age of 30 with initial CIN2 who are managed conservatively can be discharged from colposcopy once histology is normal or LSIL and HPV is negative on two consecutive annual follow-up visits (conditional, moderate).
- People under the age of 30 with initial CIN2 who remain HPV-positive at annual follow-up should remain in colposcopy with HPV tests at annual intervals (conditional, moderate).

There is controversy about the reporting of HSIL into distinct CIN2 versus CIN3 categories as there is low inter-rater reliability for CIN2 among pathologists [37]. However, due to the real obstetrical harms from over-treatment [43], there are multiple studies supporting conservative management of CIN 2 in reproductive-age people. The data for regression of CIN2 are strongest in the <30 year age group and decrease with age [44,45]. A systematic review and meta-analysis of 36 studies including 3160 people showed CIN 2 regression rates of 50% at 24 months, regardless of age [44]. For those under the age of 30, regression rates were higher and peaked at 70% at 36 months, highlighting the need

for prolonged follow-up if conservative management is undertaken. Regardless of age, regression was lower among those who were HPV16/18-positive, 21%, at 24 months. The rate of non-compliance (including loss to follow-up and missing data in retrospective studies) was 19% at 6 months, which indicates the importance of appropriate selection of candidates for conservative management [33]. Newer studies continue to support findings of high regression rates and low cancer rates with conservative therapy in CIN2 [46–49]. Recent Canadian studies from British Columbia and Nova Scotia focusing on conservative management of CIN 2 in people 24 years of age and younger showed 73–75% regression rates in 6–12 months [50,51].

The conservative management of the CIN2 pathway presented here (Figure 4) addresses people under the age of 30 with biopsy-confirmed histologic CIN2, where wishes for childbearing outweigh the risk of cancer. If conservative management is undertaken, a review by an experienced cytopathologist should exclude the presence of CIN3. In the absence of confirmed CIN3, people with a type 1 or 2 transformation zone may undergo conservative management with surveillance in colposcopy, although special consideration should be made for an excisional procedure if the person is confirmed to be HPV 16/18-positive. In people with CIN2 and a type 3 transformation zone or CIN2 on endocervical curettage, the recommendation is for an excisional procedure.

For those who are treated conservatively, colposcopy is recommended at 6-month intervals with HPV testing annually. This allows time for regression in this young cohort. At any point during follow-up, if CIN3 is found, an excisional procedure is required. Similarly, if histologic CIN2+ persists >36 months, an excisional treatment is recommended. If HPV remains positive with normal or LSIL histology, the person should remain in colposcopy including HPV testing at annual intervals. Once histology is normal or LSIL and HPV is negative on two consecutive annual follow-up visits, the person can be discharged from colposcopy to 12-month HPV-based screening with their primary care provider. Subsequent screening then follows the 'Post-discharge follow-up for people with SIL not treated in colposcopy' pathway (Figure 7).

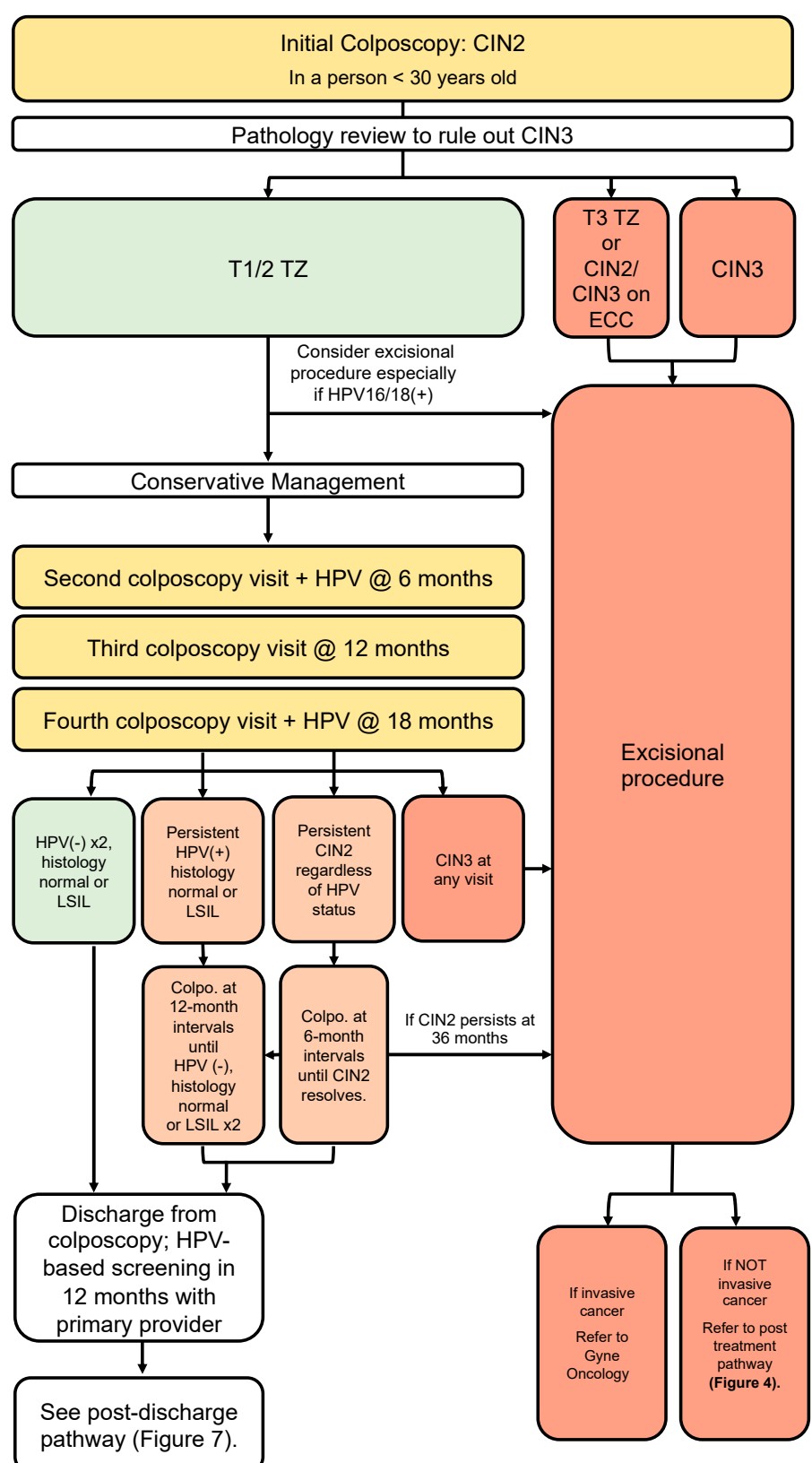

**Figure 4.** Conservative management of CIN2 in persons under the age of 30. This pathway addresses persons under the age of 30 referred to colposcopy with HPV-positive ASC-H and HSIL cytology, where wishes for childbearing outweigh risk of invasive cancer.

*3.7. Treatment for HSIL Histology*

Recommendations:

- Recommended treatment for histological HSIL is an excision procedure with a LEEP (strong, high).
- An ablative procedure with carbon dioxide laser is acceptable when used by trained and experienced colposcopists when specific criteria are met (conditional, low).
- Cryotherapy for HSIL is not recommended (strong, high).
- Treatment should be performed in the clinic setting with local anesthesia plus a vasopressor to the cervix (conditional, moderate).

Histologic HSIL is treated with either an ablative or excisional procedure. Excisional procedures offer a secondary diagnostic benefit as it allows for pathologic examination to assess for cancer and margin status. Excisional techniques include the loop electrosurgical excision procedure (LEEP), large loop excision of the transformation zone (LLETZ) and cold knife conization (CKC). Meta-analyses have shown similar oncologic outcomes, including recurrence rate, margin status and residual disease between LEEP and CKC [52,53]. However, data suggest that the rates of obstetrical complications are higher following CKC compared to LEEP [54]. For these reasons, as well as due to the logistical benefit of offering LEEP in the clinic, LEEP has become the gold-standard treatment for HSIL in Canada.

Ablation with either carbon dioxide laser or thermoablation have also been used to treat HSIL. When ablation is used, the transformation zone is destroyed to a depth of 7 mm, as a SIL can exist to a depth of at least 4 mm in gland crypts and 7 mm is considered a suitable safety margin [55]. Ablative techniques should only be used by trained and experienced colposcopists when the criteria are met (Table 4) [16,56]

**Table 4.** Criteria for ablative treatment.

| |
|---|
| The transformation zone must be fully visible (type 1); |
| The lesion should not extend to the endocervix or vagina; |
| The lesion should not occupy more than 75% of the ectocervix; |
| The transformation zone can be covered by the largest ablative probe; |
| There is no cytological/histological disparity; |
| The person has not had previous treatment; |
| There is no suspicion of cancer or glandular lesion. |

With excisional procedures, the goal of achieving negative margins must be balanced with the risk of obstetrical complication. A meta-analysis suggests cervical dysplasia is a risk for preterm birth and excisional and ablative treatments increase that risk, with increasing depths of excision/ablation increasing obstetrical risks [57]. Across several studies, margin status has been shown to be an independent risk factor for recurrent disease [58–64]. This is especially true of the endocervical margin. One study of over 3500 people reported that those with positive endocervical margins had a higher persistence rate than those with a positive ectocervical margin (13.2% (23/174 vs. 4.7% (13/278) [65].

Despite this knowledge, no study has been able to determine an optimal resection volume/length for excisional techniques. A 10 mm length seems to provide adequate margins while maintaining adequate cervical length for future pregnancies where risk of preterm birth increases with depth >10 mm [55]. Treatment should be individualized where deeper excisions are more likely required for type 2/3 transformation zones, in older people who have completed childbearing and for glandular lesions. For example, a 15–20 mm excision may be required to treat a lesion in a type 3 transformation zone [56]. Very little data exist to support "top-hat" LEEP of the endocervix for squamous lesions and thus they are not generally recommended [66].

There are very few randomized controlled trials that directly compare treatment modalities. Most of the evidence comes from systematic reviews and meta-analyses with high heterogeneity. Minor and major adverse events from any treatment are low across the studies (<1%) [57]. The most common risks of LEEP are minor bleeding (2.4%) and

infection, with rarer events being preterm labor (where risk depends on length of excision) and cervical stenosis [57].

Analgesia is required for any ablative or excisional treatments of the lower genital tract. The most accepted technique is para-cervical block with local anesthetic and a vasopressor. The vasopressor reduces blood loss, reduces the threshold of toxicity from the local anesthetic and has been shown to reduce pain compared to local anesthetic alone. A systematic review and meta-analysis by Gajjar et al. confirmed that intracervical injection of local anesthetic with a vasopressor provided optimum analgesia for treatment, although data were scant, underpowered and showed a high degree of variability [58]. Optimal dosing regimens are unclear. Importantly, the addition of topical or oral analgesics did not appear to improve pain scores. Nonetheless, oral analgesics continue to be a reasonable adjunct to para-cervical block for treatment. These recommendations are in line with the recommended best practices document provided by Cancer Care Ontario in 2016 [59].

It is recommended that LEEP is performed in the clinic setting. The benefits of treatment in the outpatient clinic, as opposed to the operating room, include faster access to treatment, quicker recovery and overall cost-effectiveness. A LEEP in the operating room or under general anesthesia should only be carried out in extenuating circumstances [59]. It is recommended that colposcopy clinics use the number of LEEPs performed in the OR versus the clinic setting as a quality indicator, with a target of at least 80% of LEEPs being performed in the clinic [60].

### 3.8. Post-Treatment Pathway (Figure 5)

Recommendations:

- All people undergoing an excisional procedure in colposcopy should have an HPV test of cure and cytology at 6 months post treatment, as well as colposcopic assessment and endocervical curettage if endocervical margin is positive on excision specimen (strong, high).
- People who are HPV-negative with normal, ASCUS or LSIL cytology and histology after treatment can be discharged from colposcopy (strong, moderate).
- People who remain persistently HPV-positive with cytology and/or histology that is normal, ASCUS or LSIL, regardless of genotype, should remain in colposcopy at 12-month intervals until HPV is negative (conditional, low).
- After discharge from colposcopy, people treated for HSIL should have 12-month HPV-based screening with their primary care provider. If HPV-negative, they can resume HPV-based screening at 3-year intervals indefinitely (conditional, low).

The post-treatment pathway (Figure 5) addresses all people undergoing excisional procedure in colposcopy where no cancer was found. Those undergoing excisional procedure should have an HPV test of cure and cytology at 6 months post treatment, as well as colposcopic assessment and endocervical curettage if the endocervical margin is positive on excision. Two recent systematic reviews have shown HPV with a cytology co-test to be more sensitive and specific than HPV or cytology alone in determining the risk of HSIL post treatment [61,62]. Furthermore, strategies that incorporate HPV testing post treatment have been shown to be safe and cost-effective in the Canadian context by reducing subsequent colposcopy visits and repeat treatments without impacting cervical cancer incidence and mortality rates [63].

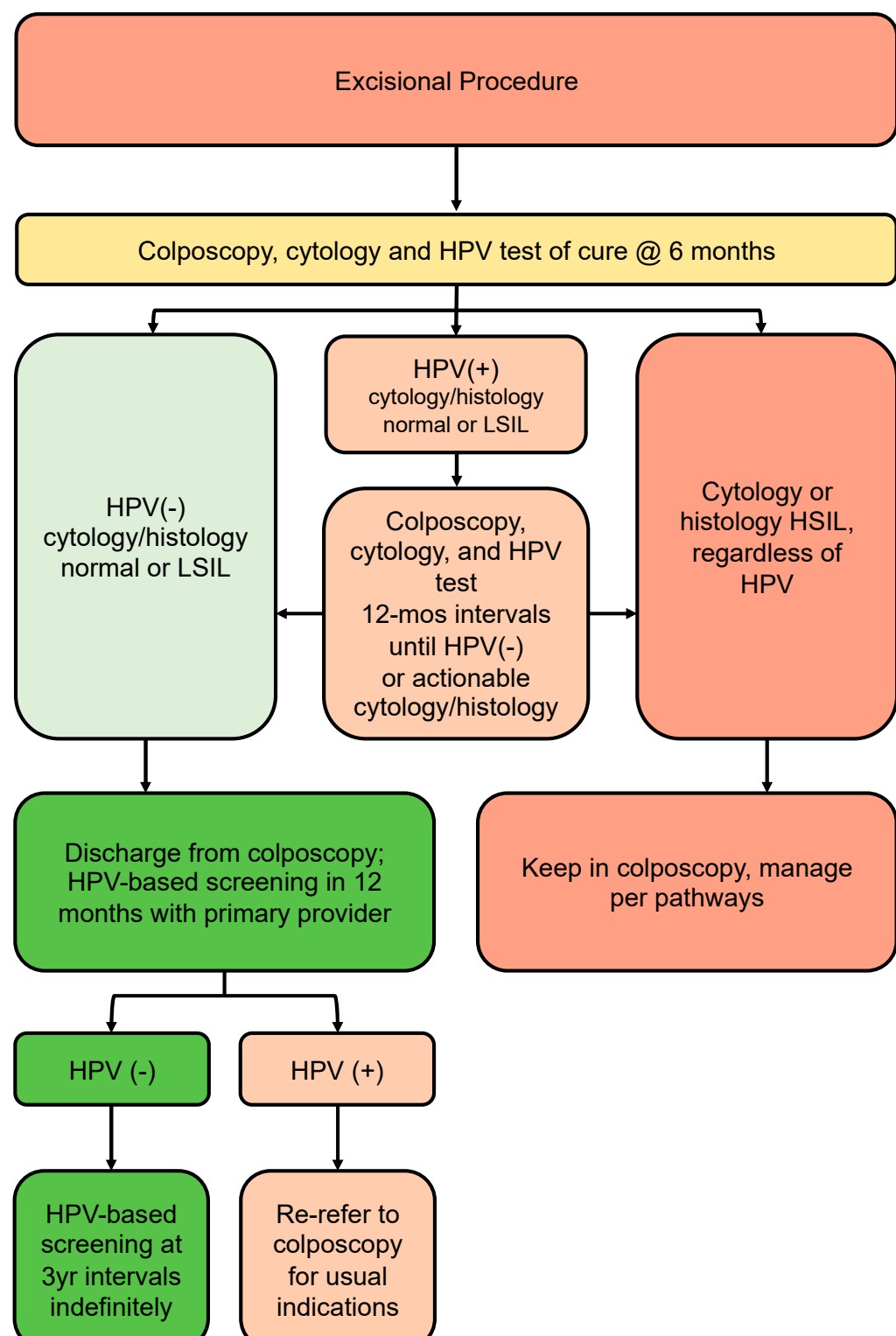

**Figure 5.** Post-treatment pathway (no invasive cancer identified). The post-treatment pathway addresses all persons undergoing excisional procedure in colposcopy where no invasive cancer was found.

A recent review of guidelines from seven countries (the USA, Denmark, Norway, Australia, the United Kingdom, Sweden and Finland) showed no agreement regarding timing of the test of cure and whether one or two post-treatment tests of cure are needed to decide who can be discharged from colposcopy [64] A study by Katki et al. of over 3000 people treated for HSIL or AIS showed that 5-year recurrence risk of HSIL+ was lower after one negative post-treatment co-test (HPV plus cytology-2.4%) compared to one negative HPV test (3.7%) or negative cytology (4.2%) alone [67]. Two consecutive co-tests did not significantly reduce the risk of recurrent HSIL further [67]. This 2.4% recurrence risk is below the colposcopy threshold of 4% determined by the 2019 ASCCP guidelines, and the 5% threshold used for this guideline, but above the return to routine 5-year screening threshold of <0.15% as outlined by the ASCCP. Furthermore, among those treated for histologic HSIL, risk of HSIL+ appears to be elevated above population risk for at least 25 years after treatment, regardless of further screening results, and therefore should be screened at more frequent intervals upon discharge from colposcopy [68–70].

Based on recurrence risk, people treated in colposcopy who are HPV-negative with normal, ASCUS or LSIL cytology and histology at the 6-month post-treatment visit can be discharged from colposcopy. They should undergo HPV-based screening with their primary care provider at 12 months post-colposcopy; if HPV is negative, they can transition to HPV-based screening at 3-year intervals indefinitely. At any point, if HPV is positive, they should be re-referred to colposcopy for the usual indications. People who are HPV-positive on the 6-month test of cure should remain in colposcopy with HPV testing annually until negative.

### 3.9. Glandular Pathway (Figure 6)

Recommendations:

- All people with HPV-positive AGC or AIS cytology should be referred directly to colposcopy, regardless of HPV genotype (strong, high).
- At time of initial colposcopic assessment, all people with HPV-positive AGC or AIS should have endocervical curettage (strong, high).
- Endometrial biopsy is recommended for all people >35 years old AND/OR risk factors for endometrial cancer AND/OR atypical endometrial cells on cytology (strong, moderate).
- People referred with AGC-NOS cytology, in the absence of endometrial pathology and histologic HSIL or AIS, may be managed conservatively with colposcopy, cytology and HPV testing at annual intervals. If all negative on two consecutive visits, they can be discharged from colposcopy to HPV-based screening at 5-year intervals (conditional, low).
- All people AGC-N or AIS should undergo excisional procedure regardless of HPV status (strong, high).
- Following the excisional procedure for AIS/AGC-N, if margins are positive, consider re-excision (strong, high).
- Following excisional procedure for AIS/AGC-N, in the absence of cancer, surveillance with 6-monthly colposcopy, ECC and HPV testing is recommended; if surveillance is negative ×3 years in colposcopy (HPV-negative, ≤HSIL/AIS), people can be discharged to HPV-based screening at 3-year intervals. If HPV is persistently positive or histology shows HSIL or persistent glandular abnormalities, people should stay in colposcopy and be managed per algorithms (conditional, moderate).
- Hysterectomy can be considered when post-treatment margins/ECC are persistently positive for AIS and/or fertility is not desired (conditional, moderate).

Colposcopy is less reliable for the assessment of cervical glandular lesions than for squamous intraepithelial lesions. Most glandular lesions are recognized during evaluation of abnormal squamous cytology and small lesions that originate high in the endocervical canal can easily be missed [71]. However, most glandular lesions occur close to the transformation zone [72,73]. Colposcopic features of glandular lesions (AIS or adenocarcinoma)

are challenging to identify and often overlap with those of squamous lesions, immature squamous metaplasia and other mimickers, highlighting the importance of tissue biopsy to confirm diagnosis [74].

Although atypical glandular cells (AGC) make up <1% of cervical cytology samples, approximately one-third of these are associated with clinically significant pre-invasive or invasive disease [75–79]. In a Canadian report, 456 cases of AGC were identified from a database of over 1 million Pap smears (0.043%) [80]. On final histologic follow-up, 7% were found to have LSIL, 36% had HSIL, 20% had AIS and 9% had cervical cancer; endometrial pathology was found in 29%, including carcinoma of the endometrium in 10% [80].

Risk estimates for HPV-positive glandular lesions are dependent on the cytology classification, as described by the Bethesda system in 2014 (Table 5). A large population database from Northern California estimated the immediate risk of HSIL+ among people with HPV positive AGC (all categories) to be 26%; this risk is substantially higher for AGC-favor neoplasia and adenocarcinoma cytology at 55% [5].

**Table 5.** 2014 Bethesda system for reporting of cervical glandular lesions [81].

| |
|---|
| Atypical Endocervical cells, not otherwise specified (-NOS) |
| Atypical Endometrial cells, not otherwise specified (-NOS) |
| Atypical Glandular cells, not otherwise specified (-NOS) |
| Atypical Endocervical cells, favoring neoplastic (-N) |
| Atypical Glandular cells, favoring neoplastic (-N) |
| Endocervical adenocarcinoma in situ (AIS) |
| Adenocarcinoma—endocervical |
| Adenocarcinoma—endometrial |
| Adenocarcinoma—extrauterine |
| Adenocarcinoma—not otherwise specified (NOS) |

The glandular referral pathway (Figure 6) addresses people referred to colposcopy with HPV-positive glandular cytology including AGC and AIS. All people with AGC and AIS cytology should have colposcopy +/− directed cervical biopsies, and endocervical sampling, regardless of HPV genotype [39]. Endometrial sampling is indicated for all people aged 35 or older, those with abnormal bleeding or other risk factors for endometrial cancer and all people with atypical endometrial cells on cytology [5,16,22,42]. Further management is based on referral cytology. For people with AGC-NOS referral cytology, management depends on colposcopy findings. For those where colposcopy findings identify normality or LSIL, surveillance is recommended. These people can be followed annually with colposcopy at 6-month intervals, including HPV testing at 6 and 18 months. If HPV remains negative and cytology is normal, ASCUS or LSIL for two consecutive annual visits, people can be discharged from colposcopy to 12-month HPV testing with their primary care provider. If HPV remains negative at 12-month post-colposcopy screening, people can resume routine HPV-based screening at 5-year intervals. If at any point HSIL or AIS are identified, they should follow the appropriate pathways for management.

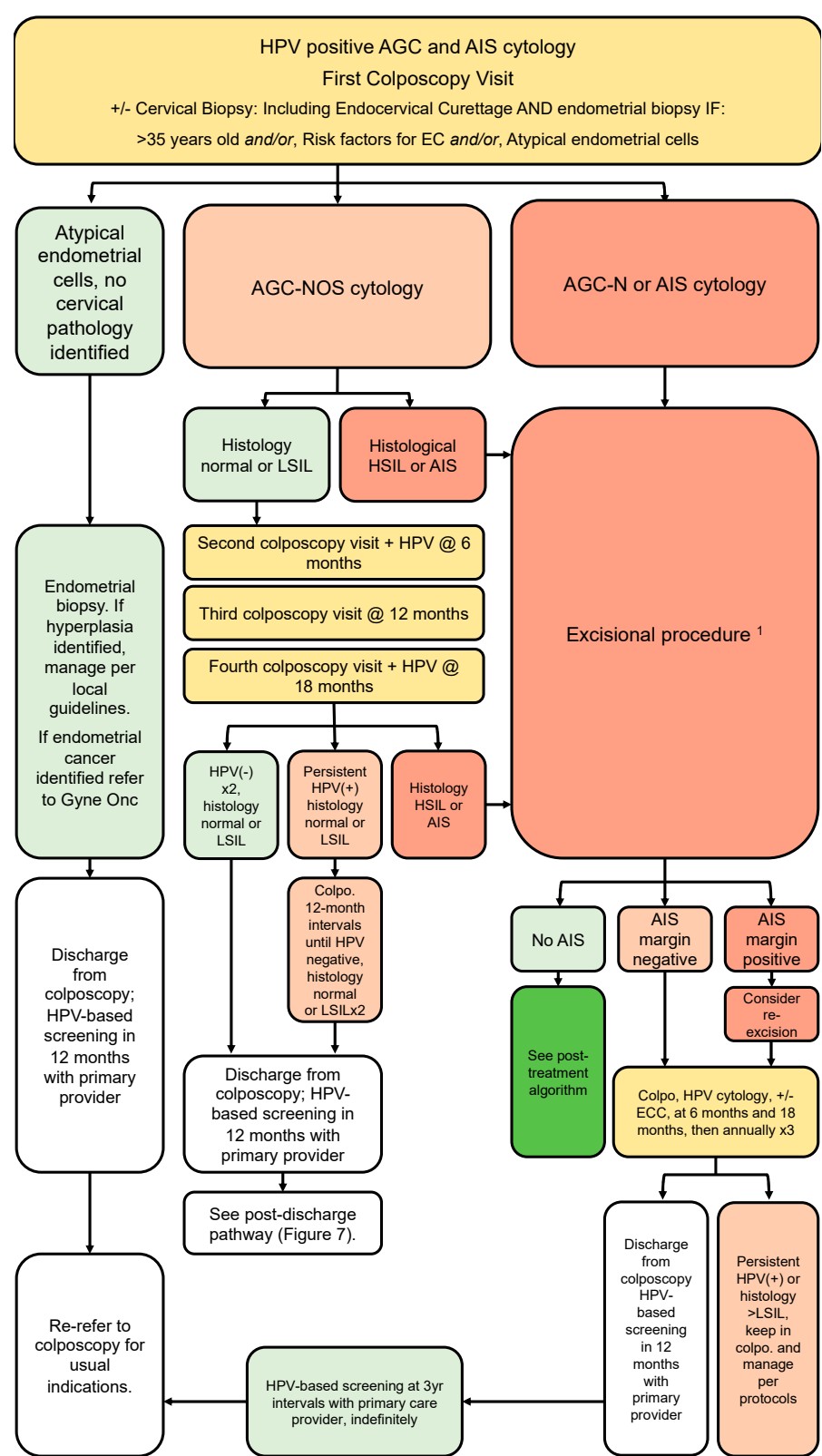

<sup>1</sup> If invasive cancer, refer to Gyne Onc.

**Figure 6.** Glandular referral pathway (HPV-positive AGC and AIS cytology). The glandular pathway addresses persons referred to colposcopy with HPV-positive AGC or AIS cytology. Initial workup should include endocervical curettage and endometrial biopsy where indicated. Further management is based on referral cytology.

For people referred to colposcopy for HPV-positive AGC-N who present with a type 3 transformation zone, a diagnostic excisional procedure is required to rule out a glandular lesion not identified at colposcopy. A systematic review by Schnatz et al. found the risk of AIS and malignancy following AGC-N cytology was 13% and 21%, respectively, compared to 2.9% and 5.2% for AGC-NOS [82]. Therefore, for people referred with AGC-N cytology, excisional procedure is mandatory. If no AIS is identified on excision specimen, the person should follow the post-treatment algorithm for subsequent management (Section 3.8; Figure 5).

### 3.9.1. Adenocarcinoma In Situ

All people with cytology suggestive of adenocarcinoma in situ (AIS) should undergo a diagnostic excisional procedure to rule out adenocarcinoma, even when definitive hysterectomy is planned [16,42,74]. The majority of AIS lesions and early adenocarcinomas are contiguous with the squamocolumnar junction; therefore, a cylindrical excision of the entire SCJ should be performed. The excision should measure at least 15 mm in length (type 3 excision) for AIS, with deeper excisions considered in perimenopausal/menopausal people, as the squamocolumnar junction retreats upwards and childbearing is no longer a consideration.

Historically, CKC was considered superior to LEEP for AIS. However, more recent retrospective reviews have shown equivocal rates of residual disease and recurrence between these treatment modalities [83,84]. Many guidelines now acknowledge LEEP as an acceptable treatment modality for AIS, so long as the specimen is removed intact and margins are interpretable [5,42,85]. Margin status after excision of AIS is an important predictor of residual disease. Adenocarcinoma was also more frequently associated with positive margins (5.2%) than with negative margins (0.1%) [86]. A meta-analysis of 33 studies showed that the risk of residual disease was 2.6% with negative margins and 19.4% with positive margins [87]. 'Top hat' excisions for AIS are not recommended as they can interfere with interpretation of margin status [5]. However, endocervical sampling immediately following the LEEP procedure is recommended, as it has been shown to predict residual disease [39]. If margins or ECC are positive, a second excision is required, even when hysterectomy is planned [5,85]. Among those with negative margins, when managed conservatively, the risk of subsequent invasive disease has been reported to be 0.35% [86]. Therefore, a conservative approach is acceptable, so long as margins are negative, and people are agreeable to long-term follow-up [5]. If negative margins for AIS cannot be achieved, hysterectomy is recommended [5].

### 3.9.2. Follow Up for Adenocarcinoma In Situ

Adherence to long-term surveillance following conservative management for AIS is crucial; this needs to be considered when determining best management for people with AIS. Following an excisional procedure for AIS with negative margins, when conservative management is undertaken, follow-up colposcopy with HPV testing and ECC should be carried out at 6 and 18 months, then annually for 3 further years in colposcopy. After treatment, HPV status is highly predictive of recurrent AIS. A study by Costa et al. found that HPV testing significantly predicted persistence/clearance of AIS at 6-month follow-up [87]. In this study, cytology and HPV co-testing had a negative predictive value of 88.9% at 6 months and 100% at 12 months. The predictive value of cytology alone did not reach statistical significance, highlighting the importance of HPV testing in this scenario [87]. If HPV remains negative and colposcopy and ECC are normal or LSIL at 12-month intervals for 3 years post excisional procedure, people can be discharged from colposcopy to HPV-based screening at 3-year intervals indefinitely.

*3.10. Post-Discharge Follow-Up for People with Squamous Lesions Not Treated in Colposcopy (Figure 7)*

Recommendations:

- All people discharged from colposcopy should have HPV-based screening at 12 months with their primary care provider (conditional, very low).
- Subsequent management depends on referral cytology and HPV status.
- People referred with low-grade cytology who are HPV-negative on 12-month screening with their primary care provider may transition to routine HPV-based screening at 5-year intervals (conditional, moderate).
- People referred with low-grade cytology who are HPV-positive (regardless of genotype) at 12-month screening with their primary care provider should be re-referred to colposcopy for the usual indications (conditional, moderate).
- People referred with high-grade cytology (untreated) should have two negative annual HPV tests in colposcopy with colposcopic findings that are normal or LSIL before they can be discharged to 12-month HPV-based screening with their primary care provider (conditional, low). If HPV remains negative at 12-month post-colposcopy screening, they may transition to HPV-based screening at 3-year intervals indefinitely. If HPV is positive at 12-month post-colposcopy screening, they should be re-referred to colposcopy for the usual indications (conditional, high).

The post-discharge pathway (Figure 7) addresses people discharged from colposcopy who do not undergo an excisional procedure. Even when an excisional procedure is not indicated, people referred to colposcopy with HPV-positive abnormal cytology have an elevated risk of subsequent histologic HSIL+ compared to the general population and therefore should undergo more intensive post-colposcopy screening. All people discharged from colposcopy should have 12-month HPV-based screening with their primary care provider to ensure smooth transfer of care and to reduce loss to follow-up. Subsequent screening depends on referral cytology and HPV status at discharge and should be based on objective estimates of future risk of histologic HSIL+. In the absence of histologic high-grade findings, risk can be grouped according to referral cytology with HPV-positive high-grade referral cytology imparting a significantly higher 3-year post-colposcopy risk of HSIL+ (8–10%, if colposcopy findings are normal or LSIL) compared to HPV-positive low-grade or normal cytology (around 2%) [36].

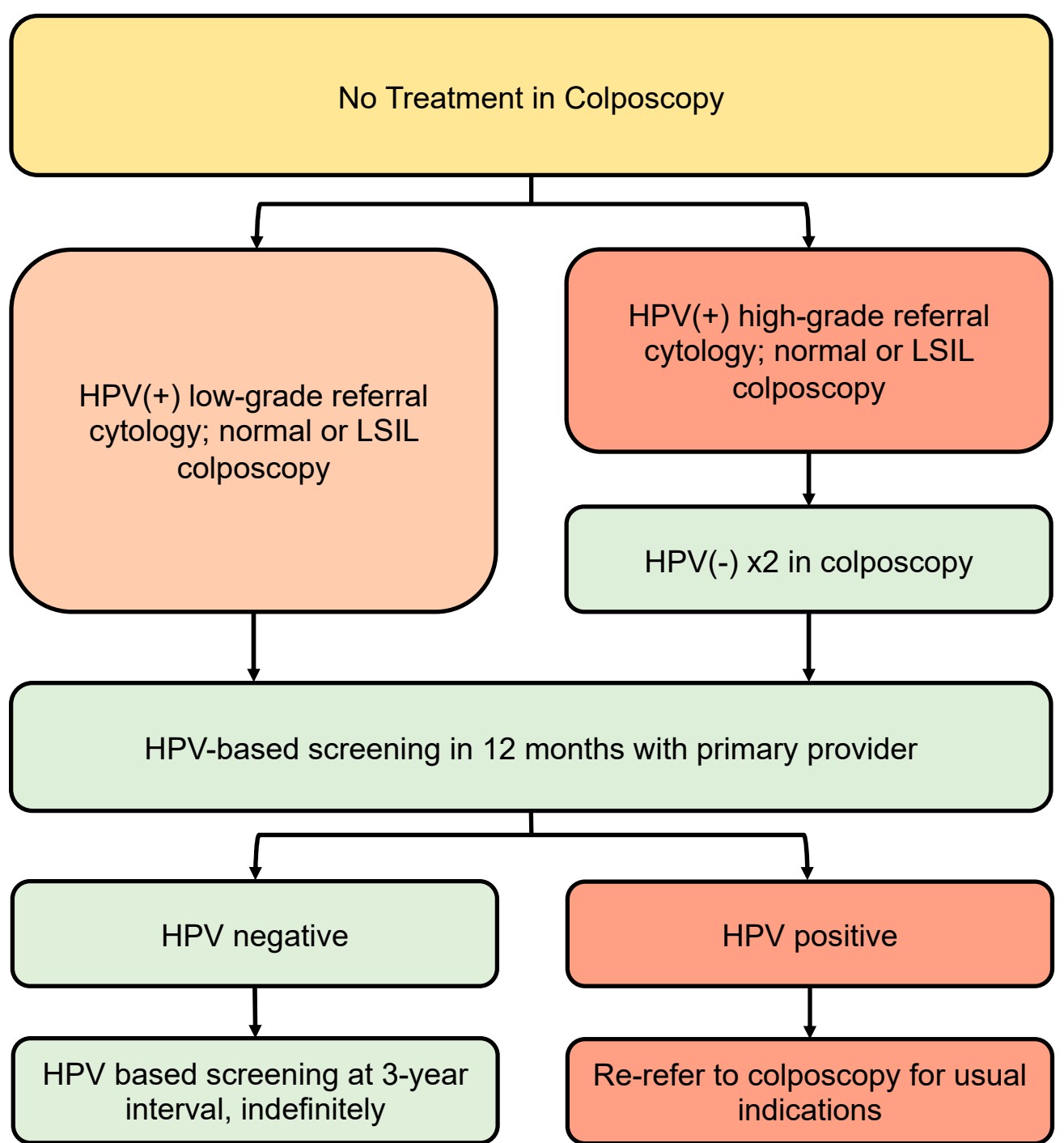

* See Figure 3 for recommendations on surveillance after untreated CIN2 in persons <30 years old. See Figure 5 for recommendations on surveillance after glandular referral/treatment for AIS.

**Figure 7.** Post-discharge follow-up for patients with SIL not treated in colposcopy *. The post-discharge pathway addresses patients discharged from colposcopy who do not undergo an excisional procedure.

3.10.1. Post-Discharge Follow-Up of People with Low-Grade Referral Cytology (Untreated)

People referred to colposcopy with HPV-positive low-grade cytology, where no histologic HSIL was found at time of colposcopy, have a low risk of subsequent HSIL within the next three years (Table 6). However, this risk does not return to baseline population risk immediately upon discharge from colposcopy, and close interval HPV-based screening is warranted. After colposcopy, follow-up HPV at 12 months with a primary care provider informs subsequent risk of high-grade disease and is more sensitive than cytology [11,60,88]. Furthermore, HPV testing 12 months from colposcopy could avoid a significant number of follow-up colposcopies among those with no histologic HSIL identified.

**Table 6.** Three-year risk of HSIL+ in people with Normal or LSIL findings at colposcopy (adapted from DeMarco 2018 [36]).

| HPV Status at Referral | Referral Cytology | Pre-Colposcopy 3-Year Risk of HSIL+, Percent | Colposcopy Findings | Post-Colposcopy (Normal or LSIL) 3-Year Risk of HSIL+, Percent |
|---|---|---|---|---|
| HPV-positive | HSIL+ | 45.4% (43.6, 47.3) | Normal or LSIL | 9.3 (0.27, 18.3) |
| HPV-positive | ASC-H | 23.9 (22.4, 25.4) | Normal or LSIL | 6.5 (2.2, 10.8) |
| HPV-positive | AGC | 26.0 (23.3, 28.9) | Normal or LSIL | 8.0 (1.5, 14.5) |
| HPV-positive | LSIL | 4.6 (4.3, 5.0) | Normal or LSIL | 1.8 (1.1, 2.6) |
| HPV-positive | ASCUS | 5.2 (4.9, 5.4) | Normal or LSIL | 2.2 (1.6, 2.8) |
| HPV-positive | NILM | 4.5 (4.1, 4.9) | Normal or LSIL | 2.1 (1.2, 3.0) |

Regarding the choice of post-colposcopy screening test, a study by Katki et al. found that for people referred with HPV-positive ASCUS or LSIL cytology, in whom no HSIL was identified at colposcopy, a single negative HPV/cytology co-test post-colposcopy reduced the 5-year risk of HSIL+ to 1.1%, compared to a single negative HPV test (2.0%) or a single negative cytology test (6.4%). A second negative HPV/cytology result in this context did not significantly reduce the risk any further [88]. Guido et al. showed that a single HPV test at 12 months post-colposcopy had the highest sensitivity (92.2%) and lowest rate of re-referral to colposcopy; re-referral rate was higher with HPV testing at 6 months post colposcopy and with co-testing in this setting [89].

Based on these data, all people seen in colposcopy for HPV-positive low-grade cytology, in which no high-grade cytology was found at the initial visit, can be discharged to HPV-based screening with their primary care provider at 12 months post-colposcopy. If HPV is negative, they can resume routine HPV-based screening at 5-year intervals. If HPV is positive on post-discharge screening, they should be re-referred to colposcopy for the usual indications.

3.10.2. Post-Discharge Follow-Up of People with High-Grade Referral Cytology (Untreated)

Despite the absence of histologic HSIL on colposcopic-directed biopsy, people who are HPV-positive with high-grade cytology at time of referral remain at higher risk of HSIL, sufficient to warrant ongoing colposcopy per the ASCCP 2019 risk-based thresholds [61,90]. All people with a history of untreated cytologic HSIL should have two consecutive negative annual HPV tests in colposcopy with no evidence of histologic HSIL/glandular lesions before being considered for discharge from colposcopy. They can then be discharged to 12-month HPV-based screening with their primary care provider. If HPV is negative at 12-month follow-up, they can resume HPV-based screening at 3-year intervals indefinitely. If HPV is positive or abnormal cytology persists, they should be re-referred to colposcopy for the usual indications.

*3.11. Special Populations*

3.11.1. People under the Age of 25

Recommendations:

- Those under 25 should not have screening with HPV testing or cytology (strong, high).
- If screening occurs and high-grade cytologic abnormalities are identified, indications for colposcopy remain the same, regardless of age (conditional, low).
- When a CIN2 lesion is confirmed, and CIN3 is ruled out, conservative management may be undertaken when childbearing considerations outweigh the risk of invasive disease (strong, moderate).

Studies have shown the highest prevalence of HPV infection is in those under the age of 25, and this population also has the highest rate of spontaneous clearance of HPV [91,92]. Invasive cervical cancer is a rare outcome of a common infection in this age group. Therefore, it is recommended that those under age 25 do not undergo screening for cervical cancer, as the risks outweigh the benefits for most in this age group [93]. If screening occurs, management should be as above, with the exception of conservative management for CIN2 in those under 30, when the histologic differentiation between CIN2 and CIN3 is available. For more detail, see Section 3.6; Figure 4.

3.11.2. Pregnancy

Recommendations

- Risk-based threshold for entry to colposcopy are the same, regardless of pregnancy (strong, high).
- Pregnant people should be evaluated by an experienced colposcopist (strong, moderate).
- Pregnant people who are HR-HPV-positive with reflex normal or low-grade referral cytology (ASCUS or LSIL) should have HPV-based screening repeated 3 months post-partum (strong, moderate); pregnant people who are HR-HPV positive with reflex high-grade or glandular cytology (ASC-H, HSIL, AGC) should be seen in colposcopy within 4 weeks (strong, moderate).
- Endometrial biopsy and endocervical curettage are contraindicated in pregnancy. (strong, high)
- Cervical biopsies are indicated when there is a concern for HSIL or cancer; adverse obstetrical outcomes of cervical biopsies are rare (conditional, moderate).
- Excisional procedures for biopsy-proven HSIL or AIS in pregnancy can be delayed until 8–12 weeks post-partum (conditional, low).
- Biopsy-proven carcinoma in pregnancy should be referred urgently to gynecologic oncology (strong, high).

Pregnant people should be screened according to provincial guidelines. Risk-based indications for colposcopy are the same, regardless of pregnancy. Pregnant people who are HR-HPV-positive with reflex normal or low-grade referral cytology (ASCUS or LSIL) should have HPV-based screening repeated 3 months post-partum; pregnant people who are HR-HPV-positive with reflex high-grade or glandular cytology (ASC-H, HSIL, AGC) should be seen in colposcopy within 4 weeks.

The pregnant cervix undergoes changes in appearance that may complicate colposcopic diagnosis; therefore, pregnant individuals should be evaluated by an experienced colpsocopist [94,95]. Pregnant individuals with cervical dysplasia may undergo spontaneous regression or persistence of lesions at similar rates to non-pregnant individuals. Rates of progression from cervical dysplasia to cervical cancer during pregnancy are low [94,96–99].

Endometrial biopsy and endocervical curettage are contraindicated in pregnancy as they may negatively impact the pregnancy. Cervical biopsies are indicated when there is a concern for HSIL or cancer [97–100]. The decision to proceed with a biopsy during pregnancy should be a shared decision with the individual and the colposcopist. The most frequent complication of a cervical biopsy during pregnancy is bleeding; adverse outcomes

such as miscarriage and pre-term delivery are rare [97,98]. Due to low rates of progression to cancer, diagnostic excisional procedures following biopsy-proven HSIL or AIS can be delayed until 8–12 weeks after birth [97,100]. Biopsy-proven carcinoma or micro-invasive carcinoma should be referred directly to a gynecologic oncologist.

### 3.11.3. Immunocompromised People

Recommendations:

- Colposcopy is recommended for all immunocompromised people who are HPV-positive, regardless of HPV genotype (conditional, low).

In immunocompromised people of any age, colposcopy is recommended for all cytology results if HPV-positive [5]. Immunocompromised states with increased risk of cervical dysplasia include those with: HIV; solid organ transplants; hematopoietic stem cell transplants, especially if concomitant with graft-vs.-host disease (GVHD); inflammatory bowel disease and rheumatoid arthritis if on an immunosuppressive agent; and systemic lupus erythematous regardless of therapy [13]. Risk-based estimates of development of histologic HSIL are currently lacking in this population. It is recommended that the management pathways above are followed; clinical judgement and individualized care is warranted until further data are available.

### 3.11.4. Menopausal People

Recommendations:

- Menopausal people have higher rates of cervix cancer and unsatisfactory colposcopy. Consider ECC and larger excisions when indicated (conditional, moderate).
- Consider pre-treatment with vaginal estrogen for 6 weeks prior to colposcopy to increase the rates of a satisfactory exam in postmenopausal people (conditional, low).

Colposcopy can be challenging in menopause as the transformation zone recedes up the endocervical canal (type 3 transformation zone). Abnormal lesions are more likely to involve the cervical canal and vagina [101]. On systematic review, colposcopic biopsies have been shown to be less accurate after age 50, in menopause and with type 3 transformation zones [18]. Small studies have shown that a six-week trial of vaginal estrogen prior to colposcopy can improve the rate of an adequate colposcopic exam [102,103]. ECC is more often required in menopause and should be considered in all people over age 45 with HPV 16 [20]. There is a higher risk of positive endocervical margins at LEEP [104]. Cervical stenosis post-LEEP is also much higher in postmenopausal people and there are minimal data on how to prevent stenosis in this population [105,106]. Menopausal people should continue with cervical screening until they have met the criteria for discontinuation based on local guidelines. A population database study in the US showed that, after correction for hysterectomy, incidence of cervical cancer continued to rise up to the age of 70 and did not start to decline until after the age of 85 [107]. In this study, 24% of people between ages 66–70 had been inadequately screened during the study period [107].

### 3.12. Equity in Colposcopy

Recommendations:

- Colposcopy providers should be aware of the barriers to access cervical cancer screening and colposcopy, including geographical, socioeconomic, cultural, physical, psychological, provider-related and system-related barriers (strong, low).
- Colposcopy providers are encouraged to seek additional training in cultural safety and trauma-informed care (strong, low).
- Every effort should be made to facilitate access to care for individuals from historically underserved populations, including people with mobility restrictions, obesity, members of the transgender community, immigrants, Indigenous peoples, people from rural communities and those with mental health disorders (strong, low).

Barriers to accessing cervical cancer screening and colposcopy can be geographical, socioeconomic, cultural, physical, psychological, provider-related and system-related. Data support inequitable access to primary cervical screening for wheelchair users [108], sex workers [109], people with obesity [110–112], members of the LGBTQ2S+ community [113–116], immigrants and refugees [117], Indigenous peoples [118–120], rural communities [121] and individuals with mental health disorders [122]. We identified a paucity of literature addressing access to colposcopy for underserved populations. Health promotion, culturally sensitive communication and gender identity inclusiveness are important to ensure more equitable access to cervical cancer screening and colposcopy. Themes of stigma, discrimination and personal history of trauma are prevalent in the literature on under-screened populations. Colposcopy providers are encouraged to seek additional training in cultural safety and trauma-informed care.

**Supplementary Materials:** The following supporting information can be downloaded at: https://www.mdpi.com/article/10.3390/curroncol30060431/s1, Table S1: Search Strategy—Sections 3.1–3.10; Table S2: Search strategy—Section 3.12 (Equity in colposcopy); Table S3: Grade of Recommendations and Evaluation of Quality

**Author Contributions:** Conceptualization, J.B., K.W. and A.S.; writing—original draft preparation, K.W., A.S., M.-H.A., B.J., N.J., J.N., L.P. and J.B.; writing—review and editing, K.W., A.S., M.-H.A., B.J., N.J., J.N., L.P. and J.B.; graphic design, M.I.; funding acquisition, J.B. All authors have read and agreed to the published version of the manuscript.

**Funding:** Production of these guidelines has been made possible through collaboration and financial support from the Canadian Partnership Against Cancer Corporation and Health Canada. The views expressed herein do not necessarily represent the views of Health Canada or the Canadian Partnership Against Cancer.

**Acknowledgments:** This guideline was developed by the Society of Gynecologic Oncology of Canada (GOC) and the Society of Canadian Colposcopists (SCC). We acknowledge members of the Pan-Canadian Cervical Screening Network who provided thoughtful feedback to the development of the draft, as well as Jesse Ehrlick, Project Manager, Precare and Carine Trazo, Managing Director, GOC.

**Conflicts of Interest:** A.S. is the current president of the Society of Canadian Colposcopists. K.W. is the president-elect of the Society of Canadian Colposcopists. K.W. has received speaker honoraria from Merck Canada. M.-H.A. has received consultation fees from GSK and Merck Canada. J.B. has received honoraria from GSK and Merck Canada and research support from the Canadian Cancer Society. The funders had no role in the design of the study; in the collection, analyses, or interpretation of data; in the writing of the manuscript; or in the decision to publish the results.

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
