# Peer review of "2023 Canadian Colposcopy Guideline: A Risk-Based Approach to Management and Surveillance of Cervical Dysplasia"

_curroncol, doi:10.3390/curroncol30060431_

Round 1

Reviewer 1 Report

Major comments:

(1) Line 50 and line 154: replace 'xxx'

(2) please check whether there are discrepant recommendations:

Line 139 Discrepancy in grading with line 807

Line 186 Discrepancy in grading with line 558

Line 191 Discrepancy in grading with line 566

(3) please check references for uniform formating

(4) A few suggestions to improve the clarity of the paper:

(i) Please include a table to describe the symbolic representation of the grading system.

(ii) To easier navigate through the paper a summary diagram of the discussed pathways (3.2, 3.4, 3.5, 3.6, 3.8, 3.9, 3.10, 3.11) would be helpful.

(iii) Also for the individual pathways the triggering condition could be the first step in the flow to allow for easier reference.

Author Response

Thank you for your comments:

Major comments:

(1) Line 50 and line 154: replace 'xxx'

thanks this is for the companion paper and will be done when we have reference for this document, it has been accepted for publication.

(2) please check whether there are discrepant recommendations:

Line 139 Discrepancy in grading with line 807:

modified

Line 186 Discrepancy in grading with line 558: 

We have reviewed the grading, line 186 refers to wide use of ECC, vs. line 558 ( now) 564 which is for ECC only.

Line 191 Discrepancy in grading with line 566

checked and modified.

(3) please check references for uniform formating

Thanks, we have done this, there were some issued with Mendeley.

(4) A few suggestions to improve the clarity of the paper:

(i) Please include a table to describe the symbolic representation of the grading system.

Thank you for this suggestion. This prompted use to review the grading system we were using. We were using an older GRADE system and so have modified all recommendations to the 2013 system. We have added an appendix C with the system on it.

(ii) To easier navigate through the paper a summary diagram of the discussed pathways (3.2, 3.4, 3.5, 3.6, 3.8, 3.9, 3.10, 3.11) would be helpful.

Thank you for the suggestion. The diagrams are large and so putting into one would be overly complex. we are planning a web site and other strategies to roll out the guideline which should help give a unified approach.

(iii) Also for the individual pathways the triggering condition could be the first step in the flow to allow for easier reference.

Thank you for this suggestion, these pathways have been modified.

Reviewer 2 Report

These colposcopy guidelines are a very useful tool for all gynecologists and healthcare professionals. They are easy to use and the flowcharts are simple to understand even if sometimes repetitive.

You could add a small paragraph in the Pregnancy chapter to define when to screen for cervical cancer. Usually, the right time is before you get pregnant during the preconception visit or in the first trimester of pregnancy. After this time pathologists have some difficulty interpreting the pap smear and any positive HPV test leads to a second trimester colposcopy which induces anxiety in these patients as well as difficult diagnostic interpretation by the gynecologist.

Author Response

You could add a small paragraph in the Pregnancy chapter to define when to screen for cervical cancer. Usually, the right time is before you get pregnant during the preconception visit or in the first trimester of pregnancy. After this time pathologists have some difficulty interpreting the pap smear and any positive HPV test leads to a second trimester colposcopy which induces anxiety in these patients as well as difficult diagnostic interpretation by the gynecologist.

Thanks for the comment. We noted that screening in colposcopy should be done per provincial guidelines, i.e. not routinely. Also note that these guidelines are intended for a new paradigm where HPV followed by reflex pap is the new paradigm. We have also noted that if the cytology is low grade they should be  seen post partum, i.e. avoiding colposcopy.